# Effect of Graphitization Degree of Mesocarbon Microbeads (MCMBs) on the Microstructure and Properties of MCMB-SiC Composites

**DOI:** 10.3390/ma16020541

**Published:** 2023-01-05

**Authors:** Shijie Huang, Xiumin Yao, Jialin Bai, Zhengren Huang, Xuejian Liu

**Affiliations:** 1State Key Laboratory of High Performance Ceramics and Superfine Microstructures, Shanghai Institute of Ceramics, Chinese Academy of Sciences, Shanghai 200050, China; 2Center of Materials Science and Optoelectronic Engineering, University of Chinese Academy of Sciences, Beijing 100049, China

**Keywords:** MCMB-SiC composites, graphitization degree, dry friction, lubricating film

## Abstract

Mesocarbon microbead-silicon carbide (MCMB-SiC) composites were prepared by hot-press sintering (2100 °C/40 MPa/1 h) with two different graphitized MCMBs as the second phase, which exhibited good self-lubricating properties. The effects of the graphitization degree of the MCMBs on the microstructure and properties of the composites were investigated contrastively. The results showed that the composites that added raw MCMBs with a low degree of graphitization had excellent self-sintering properties, higher densities, and better mechanical properties; by comparison, the composites that added mature MCMBs with a high degree of graphitization, which has regular and orderly lamellar structures, not only had good mechanical properties but also exhibited a lower and more stable dry friction coefficient (0.35), despite the higher wear rate (2.66 × 10^−6^ mm^3^·N^−1^·m^−1^). Large amounts of mature MCMBs were peeled off during the friction process to form a uniform and flat graphite lubricating film, which was the main reason for reducing the dry friction coefficient of the self-lubricating composites and making the friction coefficient more stable.

## 1. Introduction

With increasingly prominent energy issues and environmental problems, the nuclear power industry has developed in leaps and bounds. Silicon carbide (SiC) ceramics have gradually become one of the key materials for shaft seal rings in nuclear reactor main pumps [1,2] owing to their high hardness [3,4], high strength [5], low thermal expansion coefficient [6], wear resistance, and good resistance to neutron radiation [7]. Unfortunately, due to the lack of a lubricating medium, the high coefficient of the friction and wear rate of SiC ceramics [8] can easily result in seal failure, which limits their applications under dry friction conditions. 

Researchers have tried to add different kinds of carbon sources, including graphite [9,10,11], carbon nanotubes [12,13], carbon fiber [14,15,16,17], graphene [18,19,20,21], and so on, to prepare SiC/C composites with good self-lubricating properties. Guo et al. [10] found that the bending strength of the SiC ceramic decreased from 507 to 357 MPa, with the graphite fluoride content increasing from 0 to 7 wt.%, while the dry friction coefficient dropped from 0.4 to 0.2. Agarwal et al. [16] fabricated short carbon fiber (C_f_)-reinforced silicon carbide (SiC) composites by spark plasma sintering (SPS), and tribological tests showed that the composites with a 30 wt.% C_f_ exhibited the lowest coefficient of friction. Llorente et al. [22] found that graphene nanoplatelets (GNPs)/SiC composites containing a 20 vol.% of GNPs showed an enhanced wear resistance than monolithic SiC, with maximum improvements of ~70%. However, most carbon sources have the disadvantages of high cost [23] and easy agglomeration [24,25].

In recent years, mesocarbon microbeads (MCMBs) have attracted extensive attention as a new type of low-cost carbon material [26] and have been widely used in high-density and high-strength carbon materials [27], lithium-ion battery anode materials [28,29], and other fields, owing to their spherical lamellar structure, good self-sintering performance, and excellent electrical and thermal conductivity. MCMBs also serve as the second phase in the field of ceramic matrix composites (CMCs). Saeed Safi et al. [30,31] used the liquid silicon infiltration (LSI) method to fabricate MCMB-SiC composites with high flexural strength and good ablation resistance. Wang et al. [32,33] prepared MCMB-SiC composites with 0–30 wt.% MCMBs by the pressureless sintering (PLS) method at 2200 °C and studied the sliding tribological properties of dense MCMB-SiC composites with 30 wt.% MCMBs against the self-mated counterparts, WC and SSiC. It is noteworthy that there are various types of commercially available MCMBs. The materials separated from the bituminous matrix are called raw MCMBs [34], and the graphitized MCMBs, after a vacuum heat treatment at 2400 °C–3000 °C, are called mature MCMBs. Until now, both domestic and foreign scholars have conducted thorough research on SiC/C composites, but there are few reports on the application of MCMB-SiC composites. At the same time, the effects of different MCMBs on the microstructure and properties of MCMB-SiC composites and the associated wear mechanisms need to be further investigated. 

In this paper, two kinds of MCMBs with different graphitization degrees were added to SiC ceramics to improve the dry friction performance. The graphitization degree of the MCMBs was characterized, and the effect of the graphitization degree of the MCMBs on the MCMB-SiC composite ceramics was studied. The wear behavior and self-lubricating mechanism are discussed emphatically.

## 2. Experimental 

### 2.1. Starting Materials and Methods

Commercially available SiC powder (FCP-15, Norton Co., Lillesand, Norway) with a purity ≥ 99% and a particle size of 0.5–2 μm was used as raw material. Both the raw MCMBs (LT-PS, Rong Tan Technology Co., LTD., Xinbei, Taiwan) and mature MCMBs (C5022, Buwei Applied Material Technology Co., LTD, Shanghai, China) were black solid powder, and their performance data are shown in Table 1. Due to the low thermal diffusivity of SiC and C, it was necessary to add a small number of sintering aids to promote the densification of the composites by solid-state sintering. In this study, boron carbide (B_4_C) powder with a purity of ≥99% and a particle size of about 1.5 μm was used as a sintering aid. At first, 60.4 wt.% SiC, 30 wt.% MCMBs, and 0.6 wt.% B_4_C were added into ethanol and mixed by wet ball milling for 4 h. The grinding media was SiC balls, and the speed was set to 300 r/min. The mixed slurry was put into an electric blast drying oven, dried at 70 °C for 12 h, and then the dried powder was screened through a 100-mesh sieve. Finally, the mixed powder was put into a high-purity graphite mold and hot-pressed at 2100 °C for 1 h under a pressure of 40 MPa in an argon atmosphere. The MCMB-SiC self-lubricating composites prepared by adding the raw MCMBs and mature MCMBs were named R-MS and M-MS, respectively.

### 2.2. Characterizations

As a carbon material, the structural characteristics of MCMBs were characterized by Raman spectroscopy. The excitation wavelength of the Ar laser emitter was fixed at 532 nm, and the emission range was 200–3500 cm^−1^. The phase constituents of the MCMB-SiC composites were analyzed by XRD (Rigaku, D/max 2550V, Tokyo, Japan), and the layer spacing of the graphite crystals was calculated by Bragg’s Formula (1) [35]:(1)$$d_{002}=\frac{\lambda }{2\sin\theta_{002}}$$

The degree of graphitization (g) could be evaluated according to the Mering and Marie Formula (2) [36]:(2)$$g=\frac{0.3440-d_{002}}{0.3440-0.3354}$$

The ideal graphite layer spacing is 0.3354 nm, the non-graphite carbon layer spacing is 0.3440 nm, and *d*_002_ is the (002) plane spacing of the graphite sample.

The bulk density of MCMB-SiC composites was measured by the Archimedes method, and the theoretical density of the composite was calculated according to the law of mixtures. The relative density was the ratio of bulk density to theoretical density. The mechanical properties of the composites were performed on a universal testing machine (Instron Co, Instron-5566, Boston, MA, USA). In detail, the three-point bending method was used to evaluate the bending strength of the MCMB-SiC composites, and the size of the samples was 3 mm × 4 mm × 36 mm. The fracture toughness was measured using the single-edge notched beam (SENB) method on the sample with a dimension of 2.5 mm × 5 mm × 30 mm, and the notch depth was 2.5 mm. The microstructure of the polished and fractured surface of MCMB-SiC composites was observed by SEM (Hitachi, SU8220, Tokyo, Japan).

Tribological tests of the composites by self-pairing counterparts were carried out on a friction testing machine (Sida, MM-T2000, Jinan, China). Each test continued for 1 h and was conducted under dry friction conditions in the air at room temperature under 49 N load and a speed of 200 rpm. The specimens were machined into a block with dimensions of 6 mm × 7 mm × 30 mm and polished away on the working surface. In the test, the self-matching counterparts were machined to Φ 40 mm × 10 mm (the inner diameter was 16 mm). Wear rates (W) were measured using a white light interferometer (Bruker, Contour GT-K, Billerica, MA, USA) and then calculated according to the following Equation (3) [37]:(3)$$W=\frac{\Delta V}{F\times S}$$
where W (mm^3^/N⋅mm) represents the wear rate; ΔV (mm^3^) represents the wear volume; F (N) represents the applied load, and S (mm) represents the sliding distance of the test.

During the test, the dry friction coefficient (μ) of the MCMB-SiC composites was automatically calculated and recorded by the tester. The worn surface was characterized by SEM (Tescan, Clara, Brno-Kohoutovice, Czech Republic, with EDS) to determine the wear mechanism. 

## 3. Results and Discussion

### 3.1. Graphitization Degree of Two MCMBs

The microstructures of the raw and mature MCMBs are shown in Figure 1. It can be seen that the raw MCMBs in Figure 1a were of good sphericity, with a disordered ordering of the internal microcrystals and small particles adhering to the surface. The raw MCMBs were prepared from asphalt, and these low molecular weight particles may be small amounts of β-resin being adsorbed on the surface of the MCMBs [38]. On the one hand, as an in situ binders during the sintering process, β-resin exhibits good plastic fluidity when subjected to pressure or heat [39]; on the other hand, the overflow of gaseous products will cause the sintering shrinkage of the MCMBs, which makes the raw MCMBs show good self-sintering performance [40]. Whereas the mature MCMBs in Figure 1b exhibited a round pie-like spherical structure, the surface tended to be flat, and the carbon layer structure was clearer and more regular. In general, after the graphitization, the specific surface area of the MCMBs decreases, the density increases, the sphericity deteriorates somewhat, and there is a reduction in the surface micropores [41].

Figure 2a displays the XRD patterns of the raw MCMBs and mature MCMBs. The low intensity and broad diffraction peaks indicated that the raw MCMBs had a low degree of graphitization and were almost amorphous. Instead, the mature MCMBs had a strong and sharp (002) graphite characteristic peak around 26° [42], which means that they had good crystallinity and reflected a high degree of graphitization. Comparing the (002) diffraction peak, it is easy to find that the diffraction peak corresponding to the (002) crystal plane of the raw MCMBs was weakly shifted towards a higher angle, and the microcrystalline layer spacing, *d*_002_, decreased. It could be observed by calculation that the graphitization degree of the raw MCMBs was 2.67%, the graphitization degree of the mature MCMBs was 90.93%, and the *d*_002_ was reduced from 0.3438 nm to 0.3362 nm.

The Raman spectra of the raw MCMBs and mature MCMBs are presented in Figure 2b. The Raman spectra of the MCMBs had D and G bands similar to the graphite near 1350 cm^−1^ and 1580 cm^−1^, which should mainly result from the defects and disordered modes in the material [43]. The G-peak position transferred from 1600 cm^−1^ of the raw MCMBs to 1580 cm^−1^ of the mature MCMBs as the graphitization degree increased. It is worth mentioning that the intensity ratio (I_D_/I_G_) of the D peak to the G peak is used to characterize the disorder degree of the carbon materials, which can reflect the existence of a large number of disordered carbon [44]. By calculation, the I_D_/I_G_ of the mature MCMBs was 0.315, which was less than 0.642 of the raw MCMBs, indicating that the order of the carbon atoms in the mature MCMBs was enhanced and more similar to the graphite. Overall, these results were consistent with the XRD analysis.

### 3.2. Microstructure and Mechanical Properties of MCMB-SiC Composites

In Figure 3, the phase constituents of the R-MS and M-MS composites are recorded. The results show that the crystalline phases of the composites mainly included α-SiC and graphite. The (002) graphite diffraction intensity of the M-MS composites was relatively higher, which is related to the higher graphitization degree of the mature MCMBs. Compared with the XRD of the initial raw MCMBs in Figure 2a, in the case of the R-MS composites, it could be found that there was a transition from a wide amorphous peak to a sharp graphite peak near the diffraction angle of 26°. Apparently, the raw MCMBs had undergone a graphitization process during the high-temperature sintering process. The degree of graphitization had been improved, and the microstructure was more orderly.

From the polished surface morphology of the MCMB-SiC composites in Figure 4, the light phase is SiC, and the dark phase is the MCMBs. Some of the MCMBs had good sphericity, while a few of the MCMBs were in the strip or an irregular shape, which can be explained by the sintering shrinkage deformation and the different orientations of the MCMBs in the matrix. Furthermore, the MCMB particles of the R-MS composites after sintering were smaller than the initial particles, which is also affected by sintering shrinkage. In contrast to the R-MS composites, due to the high graphitization degree and low hardness of the mature MCMBs, the M-MS composites are easy to fall off during processing, resulting in a more uneven polished surface. This could be associated with the different light and dark contrast of the MCMBs, as shown in Figure 4.

The cross-sectional morphology of the MCMB-SiC composites is displayed in Figure 5. As observed, both the R-MS and M-MS composites were very dense, but it should be noted that the MCMBs themselves were lamellar and porous, and there were a few pores between the carbon layers after partial shedding [45]. The interface between the SiC and MCMBs was tightly bonded, which provides evidence that the main fracture ways of the two composites are transcrystalline ruptures, while the intercrystalline ruptures are minor. The SiC was mostly fine equiaxed grains, and the main reason for the smaller grains is that the uniform distribution of the MCMBs hinders the growth of the SiC grains, which can theoretically obtain better mechanical properties. Moreover, there were more pulled-out MCMBs in the R-MS composite. It is then reasonable to think that the pull-out of the MCMBs is beneficial to deflect cracks, consume fracture energy, and improve the fracture toughness of the composites.

The relative density and mechanical properties of two MCMB-SiC self-lubricating composites are listed in Table 2. 

It is clear that the relative density, hardness, and mechanical properties of the composites decreased with the increase of the graphitization degree of the MCMBs. The relative density of the R-MS composites was 98.1 ± 0.1%, and no obvious pores were observed in Figure 5a, which means that the composites are almost completely dense. The raw MCMBs had good self-sintering properties and were, therefore, subjected to large plastic deformation by pressure during the hot-pressing sintering process. This sintering shrinkage facilitates the discharge of voids in the MCMBs, resulting in a higher density [40]. Furthermore, the M-MS composites with mature MCMBs had a lower hardness, which may decrease the wear resistance property. The current research shows that the elastic modulus is usually greatly affected by the pores [46]. The elastic modulus of the R-MS composites was also higher, reaching 174 ± 5 GPa. At the same time, the higher density and the more pulled-out MCMBs made the R-MS composites have a higher fracture toughness, which was 5.13 ± 0.27 MPa·m^1/2^. When the composites undergo fracture, the dispersively distributed MCMBs, as the second phase, are able to hinder the crack propagation or deflect the crack so as to absorb the stress and ultimately play a role in particle diffusion toughening [47,48]. Additionally, the added raw and mature MCMBs were of the same particle size, and the bending strengths of the R-MS and M-MS composites did not differ significantly, approximately 300 MPa. As a whole, the R-MS composites with raw MCMBs possess better mechanical properties.

### 3.3. Dry friction Properties of MCMB-SiC Composites

In this section, the dry friction properties of the two composites were tested by the block-on-ring method. The experimental results are demonstrated in Table 3. It can be seen that the average dry friction coefficient (μ) of the M-MS composites was 0.35. Compared with the R-MS composites, the μ of the M-MS composites was significantly reduced, but its wear rate (W) increased from 7.7 × 10^−7^ mm^3^·N^−1^·m^−1^ to 2.66 × 10^−6^ mm^3^·N^−1^·m^−1^, with an order of magnitude difference. The friction curves of Figure 6 show that the μ of the R-MS composites fluctuated periodically between 0.2 and 0.6 and then increased gradually with the extension of the sliding time. The μ of the M-MS composites remained oscillating between 0.25 and 0.5, and the μ was relatively stable throughout the friction process. Evidently, the μ of both the MCMB-SiC composites were very close at the initial stage of the test, but with the increase of the sliding time, the self-lubricating effect of the lubricating film on the surface of the R-MS composites gradually weakened. The wear resistance of the composites decreased, leading to the wear on the surface becoming more serious; thus, the μ got larger. With the addition of mature MCMBs, the M-MS composites exhibited a higher mass loss. It is speculated that these results correlate with its high porosity and low hardness. Nevertheless, the large number of fallen MCMBs contributes to the formation of a smooth graphite lubricant film with continuous and uniform coverage on the surface, resulting in a lower and more stable μ. It is notable that the wear rates of both the MCMB-SiC composites are within the level suitable for technical applications (<<10^−5^ mm^3^·N^−1^·m^−1^) [49].

Based on the microstructure of the worn surface in Figure 7 and Figure 8, the wear mechanism of the MCMB-SiC composites is further analyzed in detail. The complex wear behavior of both MCMB-SiC composites is similar, including the wear, mechanical abrasion (micro-fracture), and tribochemical reaction [50]. The microfracture zone on the worn surface is characterized by a rough surface and accumulated wear debris. As the sliding time increases, the generated frictional heat accumulates on the friction surface where the elevated temperature leads to oxidation of the surface elements, known as tribochemical reactions [51]. Comparing the surface morphology, it can be found that the friction surface of the R-MS composites demonstrated rough and shallow furrow scratches along the sliding direction (Figure 7a), while the friction surface of the M-MS composites was smooth and flat (Figure 8a). A small number of micro cracks and a slight plastic deformation appeared on the friction surface of both the MCMB-SiC composites and mainly on the lubricating film. The formation of the microcracks may be attributed to the stress generated by the applied load and friction during the test and the mismatch of thermal expansion between the surface film and the substrate material. During the friction process, the soft MCMBs are first worn, and the exfoliated MCMB particles spread on the worn surface under the action of friction and load to form a graphite-lubricating film. Hence, the μ of the two MCMB-SiC composites in this paper was between 0.3 and 0.5, which were smaller than that of monolithic SiC ceramics (0.5–0.7).

The EDS analysis (Figure 7d) of the R-MS composites’ worn area and unworn area reveals that the carbon content of the worn area is relatively low, and the oxygen content is higher than that in the unworn areas, which indicates that the tribochemical reaction occurs during the sliding process and the worn area formed a layer of oxide film covering the surface of the substrate. The formation of the tribo-oxidation film also had a tendency to fill the pits arising from mechanical abrasion (as shown in Figure 7c). In fact, since the self-pairing counterpart has the same phase composition and hardness as the specimen, the tribochemical film on the working surface will not be peeled off but is adsorbed on the substrate to form a lubricating film (Figure 7e and Figure 8e) with a lower shear force [52], and the possible chemical reaction is as follows:2SiC + 3O_2_ = 2SiO_2_ + 2CO(4)

By the EDS analysis of Figure 7e, it can be concluded that the main components of the oxide film included Si and O elements, which further proves that the surface is oxidized to form a SiO_2_ film on account of the accumulation of heat during the sliding process. However, the SiO_2_ film on the surface was subjected to tensile stress under friction. Considering the high fracture toughness of the R-MS composites, plastic deformation may occur on the oxide film and gradually peel off. Furthermore, the MCMBs of the R-MS composites drop less under a low load, so it is difficult to spread on the friction surface to form a complete and continuous lubricating film. As the sliding time increases, the lubricating film cannot be added in time, which increases the friction resistance, resulting in the μ gradually increasing.

As for M-MS composites, in addition to the microcracks, a small number of pits formed by the material removal on the worn surface can be seen in Figure 8b, and there is more debris on the compacted lubricating film. The EDS analysis of Figure 8c illustrates that the debris is mainly composed of a mixture of graphite and SiC particles with submicron dimensions, which suggests that the mature MCMBs with a small amount of SiC particles are pulled out and broken from the matrix during the sliding process. It could be inferred that due to the density of the M-MS composites being lower than that of the R-MS composites, the combination between the SiC and MCMBs is weak, which leads to the pull-out of the SiC grains during the sliding process. The mixing of hard SiC debris inside the carbon film will generate secondary wear, which increases the wear rate. 

Different from the R-MS composites, the whole worn surface of the M-MS composites is completely covered by a layer of lubricating film. The uniform spreading lubricating film will fill the surface depression, which is the reason for the smooth worn surface. Herein, the results of the EDS surface scanning analysis (Figure 8e) could certify that the worn surface of the M-MS composite is almost completely covered by the lubricating film. The film contains the elements C, Si, and O of the initial material, and a large number of O elements indicate that a friction oxidation process has taken place. In particular, there are many lubricating films stacked on the edge of the friction trace (Figure 8a). The lamellar mature MCMBs are exfoliated in large quantities and then cleaved, transferred, and spread into a layer of graphite lubricating film. Consequently, the lubricating film with a certain thickness will deposit along the sliding direction. Briefly speaking, the M-MS composites have a lower and more stable dry friction coefficient owing to the good self-lubricating effect of the graphite [53] and the better uniformity and greater coverage of the graphite’s lubricating film on the surface, although the wear rate has increased.

To sum up, the different degrees of graphitization of the MCMBs will make the performance of the MCMB-SiC composites different, which in turn affects the formation, spreading, and coverage of the carbon film on the worn surface during the friction process. The wear mechanism of two MCMB-SiC composites is shown in Figure 9. The wear forms of both MCMB-SiC composites include abrasive wear and oxidation wear. With the addition of raw MCMBs, the surface of the R-MS composites presents an obvious furrow, the wear mechanism is mainly plough wear, and the friction surface is partially covered by tribochemical film. The increase of the surface tribochemical film coverage might contribute to reducing the direct contact of the working face of the friction pair, bringing the friction force down, and significantly minifying the dry friction coefficient. The introduction of mature MCMBs reduces the hardness and aggravates the wear; on the other hand, a large number of MCMBs are stripped and crushed, leading to reduced wear. The two mechanisms compete with each other and work together. This study provides some guidelines to realize the application of MCMB-SiC self-lubricating composites on nuclear power mechanical seals.

## 4. Conclusions

In this study, the M-MS (with mature MCMBs) and R-MS composites (with raw MCMBs) were produced by hot-pressing sintering, and their mechanical and tribological properties were investigated. 

We found that the M-MS composites that added mature MCMBs with a high degree of graphitization exhibited lower densities (97.3 ± 0.5%) and worse mechanical properties than the R-MS composites. However, the average μ of the M-MS composites was 0.35, which was reduced by ~20.5%, indicating that the mature MCMBs play a good role in solid lubrication, and the increase of the graphitization degree of MCMBs is beneficial to minify the friction coefficient of the lubricating film. 

Although the average wear rate of the M-MS composites increased from 7.7 × 10^−7^ mm^3^·N^−1^·m^−1^ to 2.66 × 10^−6^ mm^3^·N^−1^·m^−1^, plenty of flaky MCMB debris could form a graphite lubricating film on the surface of the composites, which isolates the direct contact between the friction pairs, showing good self-lubricating performance. This work provides guidance for improving the dry friction performance of the SiC/C self-lubricating sealing materials, with a view to the early realization of large-scale industrial applications.

## Figures and Tables

**Figure 1 materials-16-00541-f001:**
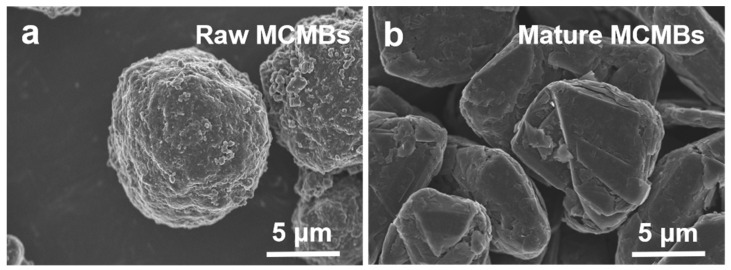
Morphology of MCMBs: (**a**) raw MCMBs; (**b**) mature MCMBs.

**Figure 2 materials-16-00541-f002:**
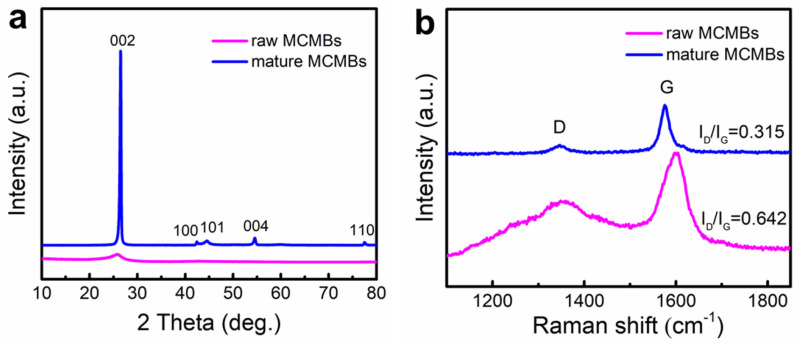
(**a**) XRD and (**b**) Raman patterns of the raw MCMBs and mature MCMBs.

**Figure 3 materials-16-00541-f003:**
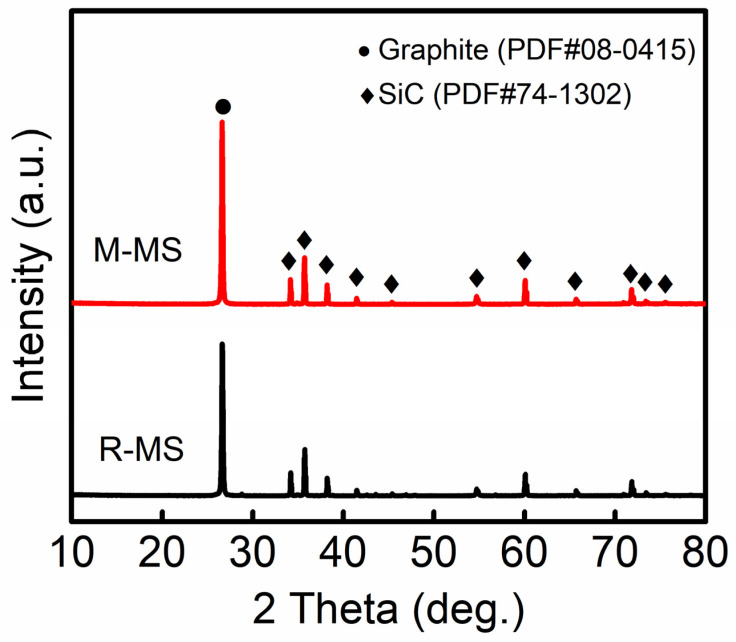
Phase constituents of MCMB-SiC composites.

**Figure 4 materials-16-00541-f004:**
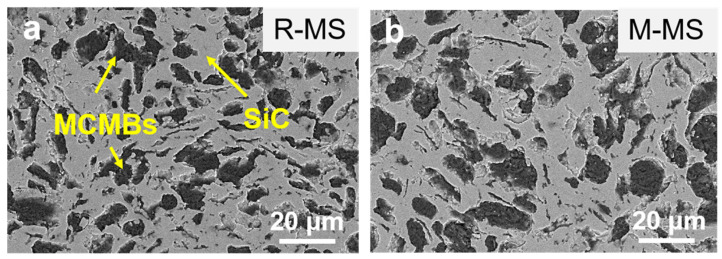
SEM images of polished surface of MCMB-SiC composites: (**a**) R-MS; (**b**) M-MS.

**Figure 5 materials-16-00541-f005:**
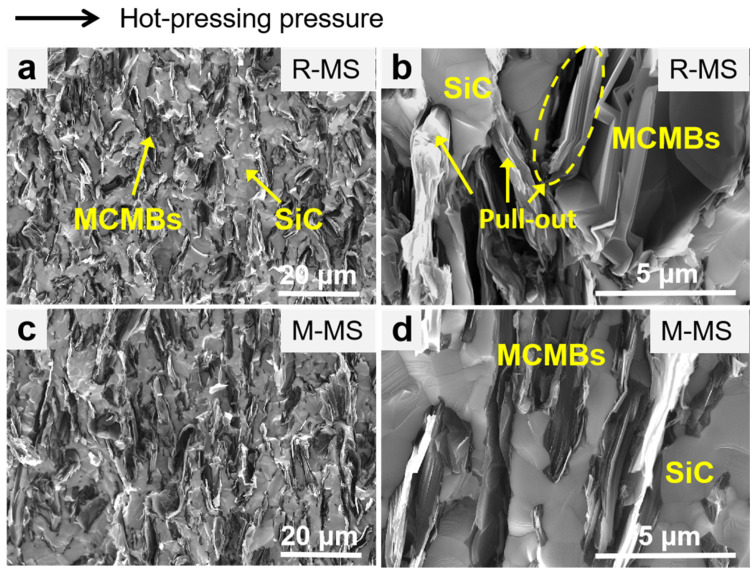
SEM images of cross-sectional of MCMB-SiC composites: (**a**,**b**) R-MS; (**c**,**d**) M-MS.

**Figure 6 materials-16-00541-f006:**
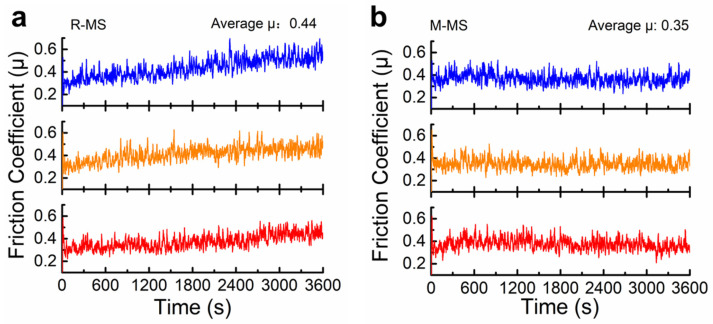
Average friction coefficient of MCMCB-SiC composites: (**a**) R-MS; (**b**) M-MS.

**Figure 7 materials-16-00541-f007:**
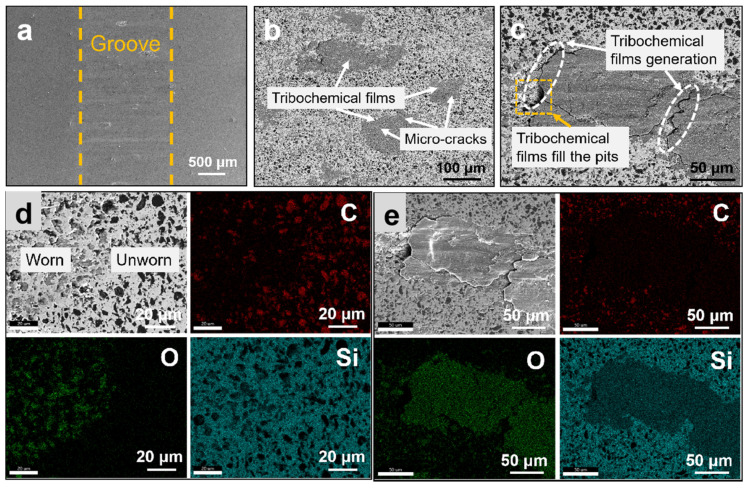
Friction surface morphology of R-MS composites: (**a**) low magnification; (**b**,**c**) high magnification; (**d**,**e**) EDS analysis.

**Figure 8 materials-16-00541-f008:**
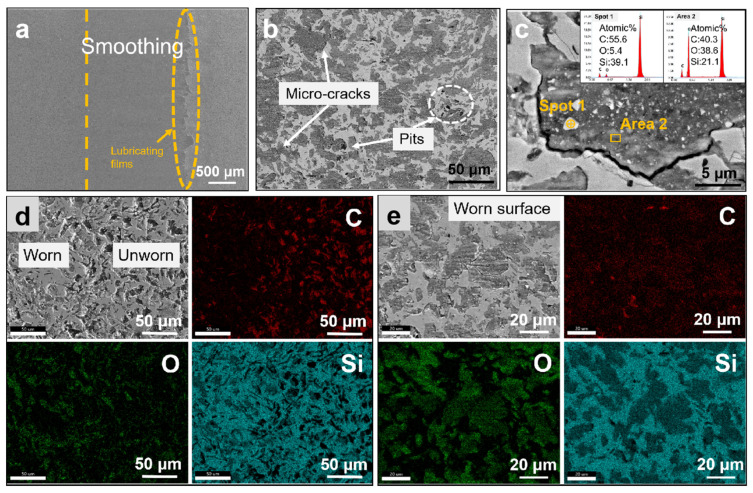
Friction surface morphology of M-MS composites: (**a**) low magnification; (**b**,**c**) high magnification; (**d**,**e**) EDS analysis.

**Figure 9 materials-16-00541-f009:**
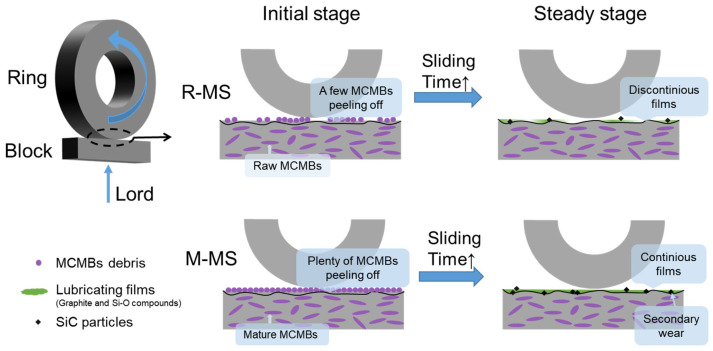
Wear mechanism of MCMB-SiC composites.

**Table 1 materials-16-00541-t001:** Performance parameters of the raw and mature MCMBs.

Material	D_50_ (μm)	Carbon Content (wt.%)	True Density (g/cm^3^)	Specific Surface Area (m^2^/g)
Raw MCMBs	10	≥99.9	≥2.2	1.2–2.2
Mature MCMBs	10	≥99.9	≥2.2	≤5

**Table 2 materials-16-00541-t002:** Mechanical Properties of MCMB-SiC Composites.

Material	Relative Density (%)	Hardness (GPa)	Flexural Strength (MPa)	Elastic Modulus (GPa)	Fracture Toughness (MPa·m^1/2^)
R-MS	98.1 ± 0.1	5.35 ± 0.43	295 ± 22	174 ± 5	5.13 ± 0.27
M-MS	97.3 ± 0.5	4.93 ± 0.44	287 ± 20	160 ± 3	4.32 ± 0.13

**Table 3 materials-16-00541-t003:** Dry friction properties of MCMB-SiC composites.

Material	Average Coefficient of Friction	Average Wear Volume (mm^3^)	Wear Rate (mm^3^·N^−1^·m^−1^)
R-MS	0.44	0.06	7.7 × 10^−7^
M-MS	0.35	0.20	2.66 × 10^−6^

## Data Availability

All the data is available within the manuscript.

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
