# Peer review of "Effect of Graphitization Degree of Mesocarbon Microbeads (MCMBs) on the Microstructure and Properties of MCMB-SiC Composites"

_materials, 2023, doi:10.3390/ma16020541_

Round 1

Reviewer 1 Report

The author did a very interesting study on Effect of graphitization degree of mesocarbon microbeads (MCMBs) on the microstructure and properties of MCMB-SiC composites but it is required to incorporate the follwoing suggestions/comments to improve the manuscript as per the standard of journal.

1. The should be more quantitative in terms of improvement.

2. The objective of work should be more elaborate with research gap.

3. Labelling the information in SEM images for better understanding.

4. It is clear that the relative density, hardness and mechan- ical properties of the composites decreased with the increase of the graphitization degree of MCMBs. Justify

5. what is the reason to increase the wear by reducing friction because it is very contradictory you can look into https://doi.org/10.1016/j.wear.2016.06.011

6. Fig 6 and 7, enlarge the text.

7. In conclusion, all results obtained by the author must be included such as mechanical properties in quantitative form.

Author Response

 Thank the Reviewer 1 for reading our paper carefully and giving the valuable

comments.

Reviewer 2 Report

This study deals with  "Effect of graphitization degree of mesocarbon microbeads (MCMBs) on the microstructure and properties of MCMB-SiC composites". The manuscript has enough innovation the result provides useful data. However, it is not well organized. Moreover, the manuscript requires further editing to conform it to correct scientific English throughout the paper. Among the areas, that need improvement is providing better citations that will help place the current work in a better context of scientific progress. many references are old and It needs a fundamental change. Moreover, the authors must check these comments:

1) Introduction section: Some sentences were not referenced.

2) The authors should show the SEM of the sample and especially the SEM of the sample for fracture toughness measurement with the SENB method (2.5 mm × 5 mm × 30 mm, and the notch depth: 2.5 mm).

3) Section, 3.1, 3.2, and 3.3: Results cannot be guaranteed in any way. The comparison of the results with other literature is very weak and requires a fundamental review in this field.

How is reaction number 4 justified? How can you be sure of its authenticity? I think there is doubt about that.

4) Conclusion section: Due to the lack of literary comparison throughout the result section, the results cannot be easily accepted in the conclusion section. Moreover, The innovation of the manuscript is not well explained. It needs to be reviewed. 

5) many references are old. Massive changes are needed in the references. 

And so on…

Author Response

Thank the Reviewer 2 for reading our paper carefully and giving the valuable

comments.

Reviewer 3 Report

This work basically studies the microstructural and mechanical properties including tribologically aspect of MCMB -SiC composites, which were prepared by hot-press sintering. The authors put a lot of emphasis on the effects of the graphitization degree of the initial MCMBs on the microstructure and other physical properties of the composites.

Looking at the background literature, two similar papers (Wang et al.; 2019; doi:10.3390/ma12193127 and Wang et al. 2020; doi.org/10.1016/j.ceramint.2019.10.116) can be found in this topic, which are already investigated the properties of MCMB -SiC system and described the tribological mechanisms of this composites in detailed. (However, one of them is missing from the reference list.) Reading these publications, I believe that this ongoing research work lacks actual novelty. Especially since this manuscript partially studies the same effect of the so-called raw MCMB on the tribological properties of the SiC composite, which was already published by above-mentioned papers. In that point of view, this “repetition” highly reduces the scientific values of this work. It is clear, that this research is focusing on the role of the different graphitisation state of the MCMB and compare the effect of the raw and mature MCMBs on the tribological properties of SiC ceramic composites, however, this is not yet an independent study.

Furthermore, based on the previous study by Wang et al. [2019], it is not clear why the authors used 30 wt% MCMB in this work? It was previously shown than 20 wt% MCMB is more effective on the tribological properties of SiC. The manuscript does not explain this question.

Apart from the above concern, the manuscript contains more weakness, and further comments raises, like:

-The comprehensive literature about MCMB-SiC is missing, and the introduction is quite poorly processed.

-The origin of the raw powders like SiC and MCMB are missing from the experimental part, as well as the synthesis process is also incomplete for the mature -MCMB. The authors only allude to that in the introduction part, like “the graphitized MCMBs after vacuum heat treatment at 2400 ℃- 3000 ℃ are called mature MCMBs”

-The obtained hardness values are around 5 GPa, which are very low despite relatively high relative density of the composites. Thus, it is supposed that the addition of MCMBs highly reduce the hardness of the SiC composites, which makes it questionable to use this composite as a seal ring. Sealing rings must not only have good tribological properties, but also be reliable in terms of hardness. Furthermore, it is written in the manuscript that „lower hardness, may decrease the wear resistance property”, which sentence is also contradictory. Thus, more precise statement is needed.

-Why cannot see the boron content on the EDX analysis? I think 0.6 wt% B4C content is quite high amount to see it by EDX.

-It would have been good if a reference sample without additives had also been tested, since a tribo-oxidation film (SiO2) is also formed in the case of additive-free SiC. Therefore, the effectiveness of MCMB would be more evidence.

-The main findings and result are not compared with other literatures values, even though they would be important for the readers.

In my opinion, this article contains significant shortcomings, therefore it cannot be accepted for publication in this journal.

Author Response

Thank the Reviewer 3 for reading our paper carefully and giving the valuable

comments.

Round 2

Reviewer 1 Report

No comments

Reviewer 2 Report

The changes are well done. Just one point,

I think the manuscript can be published with the following minor revise:

You should refer to the following references in the introduction section about SiC:

https://doi.org/10.1016/j.ijrmhm.2017.10.001

https://doi.org/10.1007/s43207-021-00173-x

https://doi.org/10.4191/kcers.2019.56.3.01

https://doi.org/10.1111/ijac.13686

Reviewer 3 Report

The authors have properly addressed all comments and suggestions from the referees. The paper, im this revised form, is ready to be accepted for publication.